# Design and Implementation of a Passive Autoranging Circuit for Hybrid FBG-PZT Photonic Current Transducer

**DOI:** 10.3390/s23010551

**Published:** 2023-01-03

**Authors:** Burhan Mir, Pawel Niewczas, Grzegorz Fusiek

**Affiliations:** Department of Electronic and Electrical Engineering, University of Strathclyde, Glasgow G1 1XW, UK

**Keywords:** fibre Bragg grating, piezoelectric transducer, photonic current transducer, current transformer, autoranging, extended dynamic range

## Abstract

In this paper, we present a novel technique for passively autoranging a photonic current transducer (PCT) that incorporates a current transformer (CT), piezoelectric transducer (PZT) and fiber Bragg grating (FBG). Due to the usage of single-mode fiber and FBG, multiple PCTs can be interconnected and distributed over a long distance, for example along a power network, greatly reducing the cost of sensor deployment and offering other unique advantages. The autoranging technique relies on the usage of multiple, serially connected CT burden resistors and associated static MOSFET switches to realize instantaneous shortening of the resistors in response to increasing measured current. This functionality is realized passively, utilizing a modular, μW-power comparator circuit that powers itself from the electrical energy supplied by the CT within a small fraction of the 50/60 Hz cycle. The resultant instantaneous changes in sensor gain will be ultimately detected by the central FBG interrogator through real-time analysis of the optical signals and will be used to apply appropriate gain scaling for each sensor. The technique will facilitate the usage of a single PCT to cover an extended dynamic range of the measurement that is required to realize a combined metering- and protection-class current sensor. This paper is limited to the description of the design process, construction, and testing of a prototype passive autoranging circuitry for integration with the PCT. The two-stage circuitry that is based on two burden resistors, 1 Ω and 10 Ω, is used to prove the concept and demonstrate the practically achievable circuit characteristics. It is shown that the circuit correctly reacts to input current threshold breaches of approximately 2 A and 20 A within a 3 ms reaction time. The circuit produces distinct voltage dips across burden resistors that will be used for signal scaling by the FBG interrogator.

## 1. Introduction

Provision of accurate voltage and current measurements in electrical power networks for metering and protection purposes is a fundamental requirement, with accuracy classes for instrument transformers specified by IEEE and IEC standards. Conventional iron-core voltage and current transformers (VTs and CTs) are the dominant technology in the energy sector for measuring voltage and current on transmission and distribution networks. However, they have significant disadvantages. Their large size and weight impact on the size of the substation and installation costs. The lack of galvanic isolation due to the usage of secondary copper leads and the risk of explosion due to the usage of oil-filled insulation systems impact on the safety of personnel. These disadvantages lead network operators to seeking alternative solutions such as the non-conventional instrument transformers (NCITs), often equipped with optical fiber communication links and digital process bus communication protocols regulated by the IEC 61850 standard, allowing for the development of compact digital substations [1,2]. 

NCITs include low-power voltage and current transformers (LPVTs and LPCTs), electronic voltage and current transformers (EVTs and ECTs) or Rogowski coils, all interfaced with corresponding electronics allowing for digital communication within substations. They provide many benefits in comparison to the traditional CTs and VTs such as reduced weight, lower environmental hazards, reduced insulation requirements, improved operational safety, lower primary circuit losses, increased bandwidth, and increased dynamic range [3,4]. However, they still suffer from the electromagnetic interference, core saturation and, fundamentally, lack passivity, requiring local power supplies [1,2].

In addition to traditional NCITs, optical sensing technologies based on the Pockels effect and fiber-optic current sensors (FOCSs), based on the Faraday effect, have been introduced in the past decades [5,6,7]. They offer several additional benefits such as light weight, small size, wide bandwidth, high accuracy, immunity to electromagnetic interference and galvanic isolation. A single FOCS can be used for DC and AC measurements with high accuracy, down to ±0.1%, covering metering and protection classes [3]. However, the commercial uptake of this technology has been slow due to the relatively high cost of the optical instrumentation solutions based on polarimetry and their vulnerability to temperature and vibration effects [7]. Moreover, optical NCITs cannot be easily multiplexed or interrogated remotely, and they are incapable of being deployed passively over long distances to offer passive wide-area network monitoring.

To overcome drawbacks of the traditional optical NCITs and to facilitate access to multiple, remote, distributed, passive current and voltage measurements over long distances, novel photonic transducers utilizing fiber Bragg grating (FBG) sensors and piezoelectric (PZT) components were proposed by the authors [8,9,10,11,12]. We previously developed an optical current sensor (OCS) for protection applications and evaluated its performance according to the industry standards. The core technology utilized in the OCS was based on a Low Voltage Transducer (LVT) employing “soft” piezoelectric components. The LVT was used directly to monitor output of a Rogowski coil or, voltage across a burden resistor connected to a conventional CT. We demonstrated that the prototype OCS was capable of meeting the accuracy requirements of the 5P class. We also developed optical voltage sensors (OVSs) for MV networks (11 kV) and HV networks (132 kV) [10,11,12]. The proposed variants utilized LVTs for sensing the secondary voltage of a capacitive voltage divider (CVD), or larger “hard” piezoelectric transducers (PZTs) measuring multi-kV-level voltages to provide direct medium voltage measurement. The devices were capable of providing voltage measurements with accuracy better than ±0.2%, showing the potential of meeting the requirements of the 3P protection and 0,2 metering classes. The proposed sensor technology can be applicable to a wide range of metering and protection applications to provide novel power system protection functions such as multi-ended circuit or multi-zone protection, or GPS-free phasor measurements that are not possible with current technology [8]. Despite its benefits, the proposed LVT-based OCS has a drawback of limited range, and it cannot compete on range or accuracy with FOCS-based NCITs. The current technology is capable of meeting either protection class (5P) or metering class (0,2) as per the IEC/IEEE standards but not class 0,2S nor both classes in one device simultaneously. As detailed in Section 3, the 0,2S class devices are required to meet stringent accuracy limits between 1% and 120% of the rated current, while for protective devices a large dynamic range for measuring currents at the rated accuracy limit is required, reaching 70 dB for combined 0,2S and 5P30 class devices.

Since the maximum current of the OCS, and hence its dynamic range, is dictated by the withstand voltage and the permissible electric field in the PZT component, whilst the interrogator noise floor limits the measurement accuracy at low current levels, a novel concept of an autoranging technique for the OCS is proposed in this paper [13]. The approach has the potential to greatly enhance the dynamic range of the sensor while offering the required measurement accuracy, making the LVT-based current sensors competitive with FOCS NCITs and offering in addition multiplexing and remote interrogation capabilities.

Consequently, the paper discusses the concept of the proposed autoranging technique and the design process, construction, and testing of a prototype passive autoranging circuitry for integration with the LVT, resulting in a novel photonic current transducer (PCT). The two-stage circuitry incorporating two burden resistors, 1 Ω and 10 Ω, is used to prove the concept and demonstrate the practically achievable circuit characteristics. It is shown that the circuit correctly reacts to input current threshold breaches of 2 A and 20 A within a 3 ms reaction time. The circuit produces distinct voltage dips across burden resistors that will be used for signal scaling by the FBG interrogator. 

## 2. Existing PCT Arrangement

A diagram of the existing PCT is shown in Figure 1 below.

The device comprises a CT, burden resistor, R_b_, current limiting resistor, R_p_, transient-voltage-suppression (TVS) diode, and low voltage transducer (LVT). The LVT is built as a separate, hermetically sealed unit. It comprises a low-voltage piezoelectric multilayer stack and a bonded fiber Bragg grating (FBG) sensor [9,14], an optical equivalent of a strain gauge. The LVT construction is shown in Figure 2. 

The FBG sensor is suspended between two ceramic arms which are attached to a cuboidal PICMA^®^ stack from Physik Instrumente (PI) [15]. The stack has nominal dimensions 5 mm × 5 mm × 18 mm and its operating voltage range is from −30 V to 120 V. The unclamped and unloaded component has a resonant frequency of 70 kHz and can reach its full displacement in approximately 4.8 μs after the driving voltage change [15].

The LVT is housed in a hermetically sealed telecommunication industry standard butterfly package, as shown in Figure 2. Voltage input to the piezoelectric stack is provided through two Kovar pins isolated from the package. Strain proportional to the input voltage is imparted to the FBG by the piezoelectric stack. The sensor can be interrogated remotely by measuring the peak wavelength reflected by the FBG, which shifts in proportion to strain and hence applied voltage due to the change in period of the grating [16]. By tracking the instantaneous peak wavelength, the voltage input can be reconstructed, and by tracking the average wavelength, local sensor temperature can be derived that can be used for temperature compensation of the sensor voltage readings [9,10,14].

The LVT construction allows for strain amplification [14], and its operating voltage range is limited to ±30 V by means of an external protection circuitry, shown in Figure 1. This consists of a protection resistor (R_p_) and a bidirectional TVS diode to avoid the piezoelectric stack depolarization and permanent damage [11]. R_p_ is used to limit the current flowing through the TVS diode during the fault conditions.

Figure 3 illustrates some potential modes of deployment of the PCT as part of the power transmission/distribution system. As depicted in Figure 3, sensors are installed at critical points of interest, for example at selected power transmission/distribution towers that may be separated by a distance of 10s of km. In this case, standard phase-to-ground fiber transition is used that acts as an insulator and conduit for the fiber which is connected at a termination box at the bottom of the pole. If fiber is not available, e.g., as part of an optical ground wire (OPGW) system, fiber can be wrapped around a phase conductor using standard wrapping technologies, such as SkyWrap [17,18,19]. Alternatively, multiple sensors can be installed at a substation, where PCTs can be mounted on insulator posts, with fiber running inside the insulator. Such an approach would allow significant benefits, eliminating large and expensive oil-filled traditional CTs, improving safety due to the inherent galvanic isolation of the PCT, and replacing multiple copper wires in the substation with a single fiber cable. 

The interrogator is then used to read and interpret the optical signals transmitted by the PCT through optical fiber. The interrogator includes a light source that produces a broad spectrum of wavelengths, a wavelength discriminating device (such as a scanning filter), and a detector that measures the intensity of the reflected light that has passed through the wavelength discriminating device. When the interrogator receives the optical signal from the PCT, it determines its peak wavelength and, on that basis, determines the magnitude of the strain applied to the FBG and hence the magnitude of the electric current being measured. The interrogator can then code the measurement data into digital sampled values according to a communication protocol for intelligent electronic devices at electrical substations, e.g., IEC61850. The data can then be transmitted to a remote location, it can be displayed, analysed, or used to control the distributed power system.

The sensors placed in remote locations of the network can be used to implement unique power system control or protection functions. For example, the sensor system may act a satellite-free distributed phasor measurement system. Another application may use the sensors to offer multi-zone differential protection. Yet, another application may involve placing the sensors at the transition points between overhead line and undergrounded cable sections to offer mixed line protection. All these applications are described in detail in [8].

## 3. PCT Dynamic Range Limitations 

According to IEC standard, 60044-8, the metering and protection classes that are routinely used by power operators are 0,2, 0,2S, and 5P (e.g., 5P30). The accuracy specifications are listed in Table 1, Table 2 and Table 3 below [20,21]. 

Based on these specifications, approximate dynamic ranges for the transducer can be calculated. This is based on the lowest and highest current values that need to be measured with the prescribed accuracy. Consequently, the dynamic ranges for the classes listed in Table 1, Table 2 and Table 3 are as follows:CT 0,2 Class: 28 dB (24/1);CT 0,2S Class: 42 dB (120/1);CT 5P30 Class: 30 dB (30/1);CT combined 0,2 and 5P30 Class: 55 dB (600/1);CT combined 0,2S and 5P30 Class: 70 dB (3000/1).

The dynamic range of the PCT is limited by the LVT maximum withstand voltage and the interrogator noise floor. The LVT maximum voltage (based on the current construction) is limited to ~20 V (rms). On the other hand, usable signals at low voltage above error levels are dictated by the IEC standards, as stated above. For the present interrogation system that is based on the I-MON 256 FBG interrogation module, metering accuracy (class 0,2) can be achieved by the PCT if the nominal voltage across the burden resistor (*I* × *R_b_*) is close to the LVT voltage limit, i.e., 20 V [11]. However, half of this limit (~10 V) will allow the device to still meet the 0,2 accuracy class and offers a buffer range. Similarly, the protection accuracy class 5P30 is achievable for nominal (*I* × *R_b_*) close to 1 V (rms). However, neither the metering class 0,2S nor the combined metering and protection class are currently directly achievable. An existing solution for range extension is to replicate the measurement chain, i.e., to have a dedicated 0,2 metering class PCT and a dedicated protection class PCT. However, this solution is still not capable to cover the 0,2S accuracy class, and it uses two sensors, increasing bulk, weight, size and cost of the solution. Additionally, the need to use two LVTs brings a disadvantage of halving the overall optical bandwidth, meaning that only half of the number of sensors can be addressed by a single interrogation unit. Currently, there is no effective solution for the PCT to cover class 0,2S or combined metering and protection class with a single device. The novel concept of PCT autoranging, as presented in this paper, will extend the measurement range without the need to replicate the voltage sensors or CTs. The concept employs multiple, serially connected burdens and static switches (e.g., power MOSFET transistors). The burdens are bypassed when the voltage across a given burden rises to a carefully selected threshold level, limiting the sensitivity of the arrangement dynamically and thus extending the measurement range of the PCT. The advantage of this approach is that the bulk and weight are kept to a minimum (only one CT and one LVT are required), the optical bandwidth is conserved, and the feature of inherent burden protection is retained as excess current is diverted from CT burden during fault or test currents. 

## 4. PCT Autoranging Circuit Concept

The concept of PCT autoranging is illustrated in the diagram in Figure 4. Here, we have an example of three burden resistors connected in series. S1, S2 and S3 are solid-state switches that are controlled by a comparator circuit that is activated when a certain CT secondary current threshold is exceeded. The switches are realized using pairs of power MOSFET transistors connected as shown at the bottom of Figure 4. If control voltage is applied between the MOSFET gate (G) and source (S) that is above the threshold conduction state for the MOSFETs, the switch will conduct. Otherwise, if G and S are shorted or the applied voltage is below the conduction threshold, the switch will be off. Note that power MOSFETs have excellent switching characteristics. They are voltage-controlled devices, with very low gate currents (~100 nA). The “off” resistance is in the range of MΩ and the “on” resistance in the range of 3 mΩ to 4.5 mΩ. Importantly, MOSFETs are capable of switching in the sub-μs range. Low-cost power MOSFETs rated to 200 A are readily available.

Example operation of the autoranging circuit shown in Figure 4.

Low current: S_1_, S_2_ and S_3_ off, allowing greatest voltage across LVT, guaranteeing the best signal-to-noise ratio.Medium current: S_1_ on, S_2_ and S_3_ off, limiting the voltage across LVT, preventing LVT damage.High current: S_1_ and S_2_ on, S_3_ off, further limiting the excess voltage across LVT and preventing its damage.Abnormal fault or test current: All switches on to eliminate overheating of the burden resistors.∙Sudden changes in optical signal due to the rapid MOSFET switching are detected by the FBG/PCT interrogator and output adjusted accordingly.

Note that a different number of resistors and associated switches can be used. For example, two resistors will enable a circuit with two threshold levels, while more than two resistors and associated switches will enable more gradual range extension.

## 5. Proof-of-Concept Design of a PCT Autoranging Circuit 

As explained above, the autoranging circuit is comprised of two (or more) burden resistors with dedicated MOSFET switches and control circuits. Moreover, the MOSFET control circuits can be replicated to actuate the relevant MOSFET pairs. To realize a proof-of-concept technology demonstrator, 1 Ω and 10 Ω burden resistors are used that are connected in series. The requirement of the design is such that when the input current crosses the low current range (2 A in the present case), corresponding to 20 V across the 10 Ω burden resistor, the circuit connected to this resistor should activate, while the circuit connected to the 1 Ω burden should be inactive. For the present design case, when the input current exceeds the 20 A threshold, the second circuit should activate to protect the 1 Ω burden resistor from overheating. Note that some network operators require that the circuit must withstand the CT secondary test current of 100 A for 1 s and 50 A for 3 s for a CT with a 1 A rated nominal secondary current [22,23,24]. The flow of the former current would result in 10 kW of power dissipated in a 1 Ω burden resistor in 1 s, rendering the construction of the unit challenging. However, the autoranging circuit would conveniently short the burden resistor during the test, allowing for the use of burden resistors with greatly reduced power ratings and physical size and bulk, simplifying the construction and reducing unit cost. 

The proposed control design is implemented using two identical control circuits to drive power MOSFET pairs. A block diagram of the proposed autoranging circuit is shown in Figure 5.

Note that in Figure 5, the high current paths are shown in red whereas the low current paths are depicted in blue. 

The interrogator of the PCT devices is a key component responsible for monitoring the current flow and detecting any sudden changes in voltage at the LVT. In normal operation, the interrogator receives a largely steady, sinusoidal voltage signal from the LVT connected to the burden of PCT. However, in certain conditions, such as a fault current, the current passing through the CT may exceed a predetermined threshold level. When this occurs, the MOSFET switches in the autoranging circuit will actuate, abruptly reducing the voltage at the burden–note again the sub-μs switching rate of MOSFETs.

In the ultimate realization of the autoranging system, the interrogator would need to be equipped with a high-speed voltage sensing capability and its firmware would need to be capable of determining sudden rapid change in voltage, ∆V/∆t, at the LVT. Such a system could use advanced signal processing techniques to accurately measure the rate of change in voltage over time, allowing it to identify the actuation of the MOSFET switches through the resulting voltage reduction/increase at the burden. For example, if a network fault current were to occur and the current passing through the CT exceeded the threshold level, the MOSFET switches would actuate and the interrogator would detect the sudden voltage drop at the burden, signaling the need for the change of measurement scale. Because of the rapid switching characteristics of the MOSFET switches, the voltage reduction/increase at the LVT due to MOSFET actuation would be easily differentiated from changes in signal magnitude due to network events (faults). Note that rapidly occurring fault conditions on the network would be at least an order (or even three orders) of magnitude slower than the surges caused by MOSFET switching. Additionally, note that the CT would act as a low-pass filter, further reducing the rate of change of any network events. Thus, the proposed methodology seems plausible.

The block diagram in Figure 6 shows an example of how an interrogator can detect changes in the secondary CT current when the current exceeds a certain threshold. When this occurs, the MOSFETs in the autoranging circuit will actuate, causing a sudden drop in the corresponding voltage detected by the LVT as illustrated in the graph in Figure 7. As discussed earlier, the firmware of the interrogator would detect a sudden change in the voltage signal of the LVT manifested as a large value of ∆V/∆t when the current threshold is breached. The left vertical axis of the graph (in blue) represents the LVT voltage and the right vertical axis of the graph (in red) represents CT secondary current. It can be seen in the graph that as soon as the current exceeds a certain threshold at a an arbitrarily indicated time instant of 45 ms, the LVT voltage drops rapidly to a low level–dictated by the remaining, unshorted, burden resistance. The interrogator will respond to this sudden change in ∆V/∆t by adjusting the gain to compensate for the reduced sensitivity of the PCT. When the current level returns to nominal value, e.g., at 85 ms in Figure 7, the MOSFETs will turn off and the LVT voltage is again rapidly increased, which will be detected by the interrogator and an appropriate gain applied again to compensate for now increased PCT sensitivity. The sensitivity of the sensors will be high during the nominal current and will be low during the fault current (or generally currents above a certain threshold). 

### 5.1. Design of MOSFET Driver Circuit

The key element of the autoranging circuit is the control circuit that drives the MOSFETs according to the specified requirements. The circuit is required to produce a voltage level that will switch the MOSFETs on for the duration of at least 1 s, and no longer than 3 s (refer to the discussion above), when the current level rises above a certain threshold value. Note that this value is dependent on the secondary nominal current, the CT class, burden resistance, and the voltage withstand of the LVT. The rule is that peak voltage across the burden does not exceed LVT’s maximum withstand voltage. The maximum withstand voltage recommended by the piezo manufacturer of the employed transducer is around +30 V and −120 V, although voltages in the range of 30 V (rms) have been shown to be withstood by the LVT without any adverse effects. Consequently, for the present circuit design, a 2 A (rms) current will create a voltage of approximately 20 V across the 10 Ω resistor and 2 V across the 1 Ω resistor, total of 22 V (31 V peak) across the LVT terminals. At this current threshold, the circuit needs to extend the measurement range by shorting the 10 Ω resistor, at which point the combined voltage across the LVT terminals will drop to 2 V. This will allow for current to rise further (by up to an order of magnitude) before the second set of switches will activate.

It is also desired that the MOSFET driving voltage is free from fluctuations as much as possible to prevent any undesirable behavior of the power MOSFETs, particularly around the threshold levels. In the DC characterization of the power MOSFETs used in the circuitry, it was observed that when V_GS_ (gate-source voltage) reaches around 4 V, the MOSFETs start conducting current with minimum resistance. It was also desired that when the current goes below the threshold level, the MOSFETs should turn off completely and there should be no operating region where MOSFETs are not completely turned on or turned off. It was observed during the characterization of the MOSFETs that there is a region just before the complete conduction where drain-source resistance is neither high nor low, potentially leading to excessive heat dissipation in the MOSFET device and its damage. One of the key goals of this design is therefore to prevent MOSFETs entering this “forbidden” region of operation and to prevent the circuit from oscillations around the threshold levels. 

A block diagram of the control circuit can be seen in Figure 8. This provides an overview of the MOSFETs driver/control circuit design.

The control/driver circuitry is based on the following three building blocks:The regulator;The threshold level detector;The comparator.

The output of the comparator is used as the gate-source drive voltage, V_GS_, of the MOSFETs. To provide the V_GS_ signal, two diodes were provided to complete a full-cycle rectifier when operating with the internal (body) diodes of the MOSFET transistors, thus forming a full bridge rectifier. Standard rectifying diodes 1N4002 were used to serve this purpose. The rectified signal that is drawn directly from the diodes cannot be used as V_GS_ signal alone. This must be first refined, regulated and fed into a control circuitry to be available as the desired V_GS_ signal. To achieve this, a comparator-based Schmitt Trigger was designed that enables the MOSFETs to completely switch on or off according to the requirements stated above.

The key design challenge is the requirement to power up the control circuit from the power available across the burden resistor within the shortest possible time from the onset of a threshold current. The reason for this is that the PCT is required to accurately reproduce any fault current that may occur on the power network within a fraction of the power frequency cycle (20 ms or 16.7 ms for the 50 Hz or 60 Hz power system, respectively). Protection systems of high voltage power networks may be required to activate within 140 ms or, exceptionally, within 300 ms. However, depending on the type of protection system, e.g., distance protection, directional comparison scheme, or directional comparison blocking scheme, command pick up times by the circuit breaker may be as low as 5 ms from the fault onset [25]. Therefore, the control circuit must power on and perform a correct decision within the time shorter than 5 ms. However, as soon as the MOSFETs start conducting with minimum resistance, the voltage at the burden resistor will drop to a minimum, switching off the power supplied to the circuit. Therefore, the circuit must accumulate sufficient charge before the MOSFETs are turned on to keep the driver circuit in operation for at least the duration of the highest test current (1 s). If current at the burden resistor continues to be supplied beyond 1 s, the MOSFETs will momentarily switch off. This, in turn, will power up the voltage regulator and the cycle will repeat. The circuit will be ‘checking’ repeatedly the status of the current level through the burden in this fashion until the current is switched off or reduced below the threshold level. If a 1 s operation was proven difficult to achieve due to the charging/discharging requirements of the circuitry, a shorter than 1 s operation should still be acceptable, provided that the momentary switching off of the MOSFETs was sufficiently short as not to cause excessive heat dissipation in the burden resistor. However, the single, uninterrupted cycle must be longer than the duration of the longest fault on the power network, e.g., 300 ms, so that the current is reproduced correctly during this critical time. 

The key building blocks of the circuit shown in Figure 8 are described as follows. The voltage regulator provides the power to drive the comparator as well as the reference voltage at the inverting input of the comparator. The threshold level detector provides a signal at the non-inverting input terminal of the comparator which is then compared with the signal at the inverting terminal of the comparator to decide when to turn the comparator output high or low. The hysteresis initiator circuit initiates a hysteresis of certain level depending on the feedback resistance when the comparator turns low from high. This prevents undesirable oscillations around the threshold levels. 

#### 5.1.1. Regulator Circuit Design

The comparator is required to be powered with the V_CC_ of 10 V DC for its correct operation. Note that the comparator needs to generate sufficiently high voltage to drive the MOSFETs. The DC voltage should be free from distortion and such that it is available to the comparator to operate ideally for at least 1 s. The regulator voltage is also required to appear immediately or within 3 ms (or less–note the discussion above). An appropriate RC based regulator circuit is proposed in Figure 9 below.

As shown in Figure 8, the input of the regulator is fed through AC/DC Power Converter. The resistor and capacitor are selected for a very low time constant, keeping into consideration that enough power is available for the comparator to operate for at least 1 s. A Zener diode D_3_ is connected at the load end to set the constant voltage level for the regulator output, which in this case is 10 V.

The time constant of an RC circuit is equal to the product of the circuit resistance in Ω and the circuit capacitance in F. The RC constant is the time required to charge the capacitor, through the resistor, from an initial charge voltage of zero to approximately 63.2% of the value of an applied DC voltage. According to the circuit in Figure 9, the time constant can be expressed as follows:(1)$$\tau_{1}=R_{r1}\times {\;C}_{1}$$

If V_C1_ is the voltage across the capacitor C_1_ in volts, V_SS1_ is the steady state capacitor voltage in volts after the charging has been completed, R_r1_ is the charging resistance in ohms in series with the capacitor, C_1_ is the capacitance of the capacitor in farads, A_1_ is the initial voltage across the capacitor C_1_ when t_1_ = 0, and t_1_ is the rise time of the regulator output in seconds, the voltage across the capacitor, V_C1_ is given by (2)
(2)$$V_{C1}=A_{1}e^{-\frac{t_{1}}{R_{r1}C_{1}}}+{\;V}_{\mathrm{SS}1}$$

At t_1_ = 0, V_C1_ = 0; substituting these values in (2) results in
$$A_{1}=-V_{\mathrm{SS}1}$$

Therefore,
(3)$$t_{1}=-R_{r1}\cdot C_{1}\cdot \ln\left(\frac{V_{C1}-{\;V}_{\mathrm{SS}1}}{A_{1}}\right)$$
(4)$$t_{1}=-\tau_{1}\cdot \ln\left(1-\frac{V_{C1}}{{\;V}_{\mathrm{SS}1}}\right)$$

#### 5.1.2. Threshold Level Detector Circuit Design

A threshold level detector is proposed to generate a signal that will be fed to the non-inverting input of the comparator to decide which state the comparator output should assume, i.e., high, or low. A proposed RC threshold level detector circuit is shown in Figure 10.

If V_c2_ is the voltage across the capacitor C_2_ in volts, V_SS2_ is the steady state capacitor voltage in volts, R_th_ is the charging resistance in ohms in series with the capacitor, t_2_ is the rise time of the threshold circuit output in seconds, C_2_ is the capacitance of the capacitor in farads, $\tau_{2}$ is the time constant of capacitor C_2_ and A_2_ is the initial voltage across the capacitor C_2_ when t_2_ = 0_,_ V_C2_ is given by (5).
(5)$$V_{c2}=A_{2}e^{-\frac{t_{2}}{R_{\mathrm{th}}C_{2}}}+V_{\mathrm{SS}2}$$

At t_2_ = 0, V_C2_ = 0. Substituting these values in (5) results in
$$A_{2}=-{\;V}_{\mathrm{SS}2}$$

Therefore,
(6)$$t_{2}=-R_{\mathrm{th}}{*C}_{2}*\ln\left(\frac{V_{C2}-{\;V}_{\mathrm{SS}2}}{A_{2}}\right)$$

$\tau_{2}$, the time constant of capacitor C_2_ is given as
(7)$$\tau_{2}=R_{\mathrm{th}}\times {\;C}_{2}$$

Substituting $\tau_{2}$ in (6) and substituting $A_{2}$ with $-{\;V}_{\mathrm{SS}2}$ yields
(8)$$t_{2}=-\tau_{2}*\ln\left(1-\frac{V_{C2}}{{\;V}_{\mathrm{SS}2}}\right)\;\;$$

The threshold output should always lag the regulator output. The requirement of threshold level detector output V_IN+_ lagging the regulator output V_CC+_ is fulfilled when t_2_ > t_1_. 

#### 5.1.3. Comparator Circuit Design

A Schmitt trigger comparator has been adopted in the design of the MOSFET driver circuit. The advantage of such a comparator is that it introduces a degree of hysteresis, thus eliminating any undesirable oscillations of the comparator output around the threshold levels and hence oscillations of the MOSFET switches. Standard comparators were not able to fulfil the requirement of the design due to their high-power consumption. A low-power consumption comparator is required in order to reduce the size of the capacitor in the regulator circuit and reduce the charging current. In the circuit, TS391ILT comparator was used. The TS391ILT comparator consumes 2 mW of power. A circuit diagram of the proposed design of the Schmitt Trigger Comparator is shown in Figure 11.

Here, $V_{\mathrm{ref}}$ is the reference voltage at the inverting terminal of the comparator and is set by the Zener diode D_5_. $V_{\mathrm{ref}}=5\;V$. V_out_ is the comparator output voltage, V_IN+_ is the non-inverting input voltage of the comparator, V_IN−_ is the inverting input voltage of the comparator and A is the open-loop gain. From the op-amp definition
(9)$$V_{\mathrm{out}}=A\;\left(V_{\mathrm{IN}+}-V_{\mathrm{IN}-}\right)$$

In the circuit in Figure 11, R_pu_ is the pull up resistor (note that the op-amp has an open-collector output stage), R_fb_ is the feedback resistor, and R_dc_ is the discharging resistor of the threshold level detector circuit (see Figure 10). V_CC+_ is the output voltage of the regulator and the supply voltage of the comparator. The analysis of the circuit, when the comparator output is assumed high, leads to the following equation:(10)$$\frac{V_{\mathrm{CC}+}}{R_{\mathrm{pu}}+R_{\mathrm{fb}}+R_{\mathrm{dc}}}=\frac{V_{\mathrm{out}}}{R_{\mathrm{fb}}+R_{\mathrm{dc}}}=\frac{V_{\mathrm{IN}+}}{R_{\mathrm{dc}}}$$

V_OH_ is the output of the comparator when it is high and V_OL_ is its output when it is low,
(11)$$V_{\mathrm{OH}}=\frac{V_{\mathrm{CC}+}*(R_{\mathrm{dc}}+R_{\mathrm{fb}})}{R_{\mathrm{fb}}+R_{\mathrm{dc}}+R_{\mathrm{pu}}}\;$$

As V_CC−_ is connected to ground
(12)$$V_{\mathrm{OL}}=V_{\mathrm{CC}-}=0\;V$$

$V_{\mathrm{TH}}$ is the threshold voltage at the non-inverting input terminal to change the transition of the comparator from low to high when it crosses the $V_{\mathrm{ref}}$ level.
(13)$$V_{\mathrm{TH}}=V_{\mathrm{ref}\;}$$

$V_{\mathrm{TL}}$ is the threshold voltage at the non-inverting input terminal to change the transition of the comparator from high to low. $V_{\mathrm{TL}}$ can be calculated as
(14)$$V_{\mathrm{TL}}=\frac{V_{\mathrm{OH}}{*R}_{\mathrm{dc}}}{R_{\mathrm{fb}}+R_{\mathrm{dc}}}$$

V_HYST_ is the voltage hysteresis created by the feedback resistor that is the difference between the two threshold detector voltages for the two transitions
(15)$$V_{\mathrm{HYST}}=V_{\mathrm{TH}}-V_{\mathrm{TL}}$$

Figure 12 illustrates that if the output of the comparator V_out_ is low it will retain its state unless V_IN+_ exceeds a certain level which is V_TH_. At that instance the comparator will change its state from low to high which in this case is V_CC+,_ i.e., 10 V. If V_out_ is high and V_IN+_ is reduced, the comparator will retain the high state even if the V_TH_ level is crossed again. When V_IN+_ is reduced from V_TL,_ the comparator will change its state from high to low which in this case is 0 V and will retain its low state unless V_IN+_ again exceeds the V_ref_/V_TH_ mark. Figure 12 illustrates that a hysteresis of V_HYST_ is created in the transition of the comparator states so that the comparator output does not exhibit oscillatory behavior which would be undesirable for the MOSFETs. Additionally, it should be noted, that the use of a fast-switching comparator prevents the drive voltage at the MOSFET gate to dwell around the “forbidden range”, preventing excessive heat dissipation in the MOSFETs.

## 6. Experimental Setup to Verify Autoranging Concept

The regulator, threshold detector, and comparator circuits shown in Figure 9, Figure 10 and Figure 11 were integrated into the complete driver circuit as shown in the block diagram in Figure 8. The resulting driver circuit was then combined with the remaining parts of the autoranging circuit as depicted earlier in Figure 5, to create the test setup for experimentation, which can be seen in Figure 13. 

A component-level diagram of the autoranging circuit together with the burden resistors and power supply arrangement are shown in Figure 14. The MOSFETs used in the circuit are FDB0300N1007L, selected on the basis of their sufficient current handling, voltage withstand and low on resistance of approximately 3 mΩ. A printed circuit board (PCB) was designed and manufactured for the autoranging circuit shown in Figure 14. A programmable AC voltage source: Chroma 61511, denoted as “Chroma Source” in Figure 14, was used to emulate the CT secondary current. The Chroma Source is capable of delivering currents of up to 144 A (rms) at a maximum voltage of 150 V. A current limiting resistor was connected between the Chroma Source output and the input of the autoranging circuit. Note that the Chroma source is a voltage source, not current source. Therefore, there is an expectation that current would rapidly increase following the activation of the MOSFET switches as the overall burden resistance rapidly decreases. This is a different behavior to that experienced as if a CT was used, which would act predominantly as a current source. Nevertheless, the present arrangement is perfectly capable of demonstrating the capability and the proper function of the autoranging circuit.

As seen in Figure 14, a series of voltage/current probes were connected to the test circuit: (i) the output of the regulator, (ii) the output of the threshold level detector, (iii) reference signal, (iv) comparator output, and (v) across the burden resistor. The total input current was recorded through a current probe pico Technology TA167. The probe was capable of measuring current up to 200 A (rms), with a frequency range up to 20 kHz at a resolution of ±100 mA and at an accuracy of ±1%. The probes were connected to the National Instruments’ data acquisition card NI USB6001 that was set up to sample the signals at 16 kHz. NI DAQ Express software package was used for recording the voltage and current waveforms on the computer system. The two parts of the circuit (associated with the 10 Ω and 1 Ω burden resistors) were investigated separately due to the limitation on the number of DAQ channels. The circuit was tested at different current levels. 

The voltage/current records were used to determine the overall characteristics of the circuit along with the switching behavior of the MOSFETs. 

### 6.1. Timing Calculations of Regulator and Threshold Detector Outputs

Based on Equations (1)–(8), the rise times of the regulator and threshold detector outputs can be calculated for the 10 Ω and 1 Ω burden part of the circuit in Figure 14 separately as follows.

#### 6.1.1. Regulator Output for the 10 Ω Burden Circuit

If V_C3_ is the voltage across capacitor C_1_ in volts, V_SS3_ is the steady state capacitor voltage in volts after the charging has been completed, R_1_ is the charging resistance in ohms in series with the capacitor, C_1_ is the capacitance of the capacitor in farads, A_3_ is the initial voltage across capacitor C_1_ when t_3_ = 0, and t_3_ is the rise time of the regulator output of the 10 Ω burden circuit in seconds, the voltage across the capacitor, V_C3_ is given by (16)
(16)$$V_{C3}=A_{3}e^{-\frac{t_{3}}{R_{1}C1}}+{\;V}_{\mathrm{SS}3}$$

At t_3_ = 0, V_C3_ = 0; substituting these values in (16) results in
$$A_{3}=-{\;V}_{\mathrm{SS}3}$$

Therefore,
(17)$$t_{3}=-R_{1}\cdot C_{1}\cdot \ln\left(\frac{V_{C3}-{\;V}_{\mathrm{SS}3}}{A_{3}}\right)$$

$\tau_{3}$ the time constant of capacitor C_1_ is given as
(18)$$\tau_{3}=R_{1}\times {\;C}_{1}$$

Substituting $\tau_{3}$ in (17) results in
(19)$$t_{3}=-\tau_{3}\cdot \ln\left(1-\frac{V_{C3}}{{\;V}_{\mathrm{SS}3}}\right)$$

#### 6.1.2. Threshold Detector Output for the 10 Ω Burden Circuit

If V_C4_ is the voltage across capacitor C_2_ in volts, V_SS4_ is the steady state capacitor voltage in volts, R_4_ is the charging resistance in ohms in series with the capacitor, t_4_ is the rise time of the threshold detector output of the 10 Ω burden circuit in seconds, C_2_ is the capacitance of the capacitor in farads, $\tau_{4}$ is the time constant of capacitor C_2_ and A_4_ is the initial voltage across capacitor C_2_ when t_4_ = 0_,_ V_C4_ is given by (20).
(20)$$V_{C4}=A_{4}e^{-\frac{t_{4}}{R_{4}C_{2}}}+V_{\mathrm{SS}4}$$

At t_4_ = 0, V_C4_ = 0; substituting these values in (20) results in
$$A_{4}=-{\;V}_{\mathrm{SS}4}$$

Therefore,
(21)$$t_{4}=-R_{4}{*C}_{2}*\ln\left(\frac{V_{C4}-{\;V}_{\mathrm{SS}4}}{A_{4}}\right)\;\;$$

$\tau_{4}$–the time constant of capacitor C_2_ is given as
(22)$$\tau_{4}=R_{4}\times {\;C}_{2}$$

Substituting $\tau_{4}$ in (21) results in
(23)$$t_{4}=-\tau_{4}*\ln\left(1-\frac{V_{C4}}{{\;V}_{\mathrm{SS}4}}\right)\;\;$$

#### 6.1.3. Regulator Output for the 1 Ω Burden Circuit

If V_C5_ is the voltage across capacitor C_3_ in volts, V_SS5_ is the steady state capacitor voltage in volts after the charging has been completed, R_12_ is the charging resistance in ohms in series with the capacitor, C_3_ is the capacitance of the capacitor in farads, A_5_ is the initial voltage across capacitor C_3_ when t_5_ = 0, and t_5_ is the rise time of the regulator output of the 1 Ω burden circuit in seconds, the voltage across the capacitor, V_C5_ is given by (24).
(24)$$V_{C5}=A_{5}e^{-\frac{t_{5}}{R_{12}C_{3}}}+{\;V}_{\mathrm{SS}5}$$

At t_5_ = 0, V_C5_ = 0; substituting these values in (24) results in$$A_{5}=-{\;V}_{\mathrm{SS}5}$$

Therefore,
(25)$$t_{5}=-R_{12}\cdot C_{3}\cdot \ln\left(\frac{V_{C5}-{\;V}_{\mathrm{SS}5}}{A_{5}}\right)$$

$\tau_{5}$–the time constant of capacitor C_3_ is given as
(26)$$\tau_{5}=R_{12}\times {\;C}_{3}$$

Substituting $\tau_{5}$ in (25) results in
(27)$$t_{5}=-\tau_{5}\cdot \ln\left(1-\frac{V_{C5}}{{\;V}_{\mathrm{SS}5}}\right)$$

#### 6.1.4. Threshold Detector Output for the 1 Ω Burden Circuit

If V_c6_ is the voltage across capacitor C_4_ in volts, V_SS6_ is the steady state capacitor voltage in volts, R_15_ is the charging resistance in ohms in series with the capacitor, t_6_ is the rise time of the threshold detector output of the 1 Ω burden circuit in seconds, C_4_ is the capacitance of the capacitor in farads, $\tau_{6}$ is the time constant of capacitor C_4_ and A_6_ is the initial voltage across capacitor C_4_ when t_6_ = 0_,_ V_C6_ is given by (28).
(28)$$V_{C6}=A_{6}e^{-\frac{t6}{R_{15}C_{4}}}+V_{\mathrm{SS}6}$$

At t_6_ = 0, V_C6_ = 0; substituting these values in (28) results in
$$A_{6}=-{\;V}_{\mathrm{SS}6}$$

Therefore,
(29)$$t_{6}=-R_{15}{*C}_{4}*\ln\left(\frac{V_{C6}-{\;V}_{\mathrm{SS}6}}{A_{6}}\right)\;\;$$

$\tau_{6}$–the time constant of capacitor C_4_ is given as
(30)$$\tau_{6}=R_{15}\times {\;C}_{4}$$

Substituting $\tau_{6}$ in (29) results in
(31)$$t_{6}=-\tau_{6}*\ln\left(\frac{V_{C6}-{\;V}_{\mathrm{SS}6}}{A_{6}}\right)\;\;$$

The requirement of the threshold level detector output, V_IN+_, lagging the regulator output, V_CC+_, for both the 10 Ω burden and 1 Ω burden circuits is fulfilled when t_4_ > t_3_ and t_6_ > t_5_. All the components were selected and tested in order to fulfil the timing requirement of the circuitry.

### 6.2. Experimental Setup to Verify Autoranging Concept and Results

A series of laboratory experiments were performed in order to confirm the correct operation of the autoranging circuit. Nomenclature for the recorded waveforms and signals in the subsequent plots can be seen in Table 4 below.

#### 6.2.1. Investigations of the 10 Ω Burden Circuit

The following figures are the results of the experimental investigations for the 10 Ω burden part of the circuit conducted at 100 A (rms) input current injected through the Chroma source. The design of the circuit in Figure 14 has been tested over a current range of 0 to 100 A (rms). Note that the graphs have limited temporal resolution of 0.062 ms due to the limited sampling frequency of 16 kHz.

Figure 15 illustrates that as soon as the power source is turned on, the regulator output V_CC+_ goes to saturation at approximately 10.5 V (because of Zener diode D_3_) within 1.5 ms. The reference voltage signal V_ref_ also originates from V_CC+_ and saturates at 5.5 V (because of Zener diode D_4_) at around 1 ms. The threshold detector voltage V_th_ lags V_ref_ initially. As soon as the saturation of V_ref_ is achieved, V_th_ crosses V_ref_ mark just before 1.5 ms as can be seen in the graph in Figure 15. V_th_ after crossing V_ref_ saturates at 7.7 V (because of Zener diode D_5_). As soon as V_th_ crosses V_ref_, V_out_ of the comparator turns high and saturates at 10.5 V. V_out_ is fed to the corresponding MOSFETs pair M_1_ and M_2_ and activates them at around 1.5 ms, resulting in the shortening of the corresponding burden resistors. The rise time of current shown in Figure 15 is extremely rapid–shorter or equal to the temporal resolution of the sampling process. This results in a rapid decrease in the voltage across the burden resistor, which will be used by the interrogation system to scale the sensor output to the appropriate level. Note that the second rapid current rise takes place around 2.6 ms. This is due to the activation of the second pair of MOSFET switches, corresponding to the 1 Ω circuit.

In Figure 16, the comparator operation can be seen along with the quality of the input current from the chroma source. The graph in Figure 16 shows one complete cycle of the input current over a duration of 20 ms, i.e., a frequency of 50 Hz with an amplitude of approximately 100 A (rms). Again, rapid rises in current can be clearly seen at the points corresponding to MOSFET switching.

Figure 17, shows that after MOSFET activation, V_out_ retains its high state for a period of 3.4 s with an intermediate “dip” at 1.64 s whose duration is less than 1 ms. Such a behavior of the circuit is acceptable because in such a short time there cannot be enough power dissipation across the burden resistor to cause damage to the resistors. Additionally, it will be shown later (Section 6.2.6.) that the voltage across the LVT is within the specified limit during the intermediate dip. 

During the experimental investigations, it was observed that at the input current of approximately 2 A (rms), near the lower switching threshold, the comparator retained its low state, preventing the MOSFETs in the 10 Ω burden circuit to activate. When the input current was increased marginally above 2 A (rms), the output of the comparator for the 10 Ω burden circuit turns high after approximately 60 ms. After the MOSFET activation, comparator output remains high for approximately 3 s with an intermediate dip of 0.5 ms at 1.64 s. When the injected current level increased further beyond 2 A (rms), the actuation time of the 10 Ω burden circuit comparator also decreased and ultimately fell within 5 ms window, desirable to handle fault currents by the PCT as their inception can be rapid.

#### 6.2.2. Burden Voltage across 10 Ω Burden Circuit (MOSFETs Switched on)

The plot shown in Figure 18 was obtained from the experiments performed on the 10 Ω burden circuit at 100 A (rms) input current to investigate the effectiveness of the shorting of the 10 Ω burden resistor, R_11_, by the MOSFETs.

The plot in Figure 18 illustrates voltage across 10 Ω burden resistor R_11,_ at 100 A (rms) while the MOSFETs are switched on. Each MOSFET (M_1_, M_2_) is expected to exhibit a resistance of approximately 3 mΩ when they are completely activated. At 100 A (rms) input current, the burden voltage can be seen as 0.55V (rms). This voltage drop across the MOSFETs will need to be taken into account in the final reproduction of the input current within the sensor interrogation system.

It should be noted that the burden voltage rises initially to a peak level of above 10 V during the MOSFETs de-activation at the intermediate dip, which could not be measured directly due to the limitation of the data acquisition card (voltage measurement range of 10 V). This sudden rise of burden voltage at the intermediate dip was investigated with the help of a voltage divider and is discussed in Section 6.2.6.

#### 6.2.3. Investigations of the 1 Ω Burden Circuit 

The following figures depict the results of the experimental investigations for the 1 Ω burden circuit carried out at 100 A (rms) input current. 

The operation of the 1Ω burden part of the circuit is analogous to that of the 10 Ω burden part as can be seen in Figure 19. The graph in Figure 19 illustrates that as soon as the power from chroma source is turned on, the regulator output V_CC+_ saturates at 10.5 V (because of Zener diode D_9_) at approximately 2.3 ms. The reference voltage V_ref_ saturates at 5.5 V (because of Zener diode D_10_) at approximately 2.4 ms. The threshold detector voltage V_th_ lags V_ref_ initially as in the case of the 10 Ω burden circuit. As soon as the saturation of V_ref_ is achieved, V_th_ crosses V_ref_ at 2.4 ms. V_th_ after crossing V_ref_ saturates at 7.7 V (because of Zener diode D_11_) at approximately 2.5 ms. As soon as V_th_ crosses V_ref_, V_out_ of the comparator turns high and saturates at 10.5 V. V_out_ is fed to V_GS_ of the corresponding MOSFETs pair M_3_ and M_4_ and activates them at around 2.5 ms—causing the MOSFETs to conduct current with minimum resistance. 

The graph in Figure 20 covers one complete cycle of the input current over the duration of 20 ms, i.e., a frequency of 50 Hz with an amplitude of approximately 100 A (rms). The rapidly rising current at the MOSFET switching times is clearly visible.

Figure 21 shows that after MOSFET activation, V_out_ retains its high state for a period of 3.2 s with an intermediate dip at 1.56 s whose duration is less than 1 ms–similar to the case of the 10 Ω burden part of the circuit. This keeps the MOSFETs pair M_3_ and M_4_ activated, forcing them to conduct current with a very low resistance. The two surges in the current waveform (steep inclines) shown in Figure 19 and Figure 20 signify the precise timing of MOSFET switching for the 10 Ω and 1 Ω burden circuits. It should be noted that whilst the recording of the waveforms generated by both parts of the circuit were performed separately, they were both combined and functioned simultaneously as evident from the current surges indicated by the current graph. 

#### 6.2.4. Burden Voltage across the 1 Ω Burden Circuit (MOSFETs on)

The waveforms shown in Figure 22 were obtained from the experiments performed on the 1 Ω burden circuit at 100 A (rms) input current to again investigate the effectiveness of the shorting of the 1 Ω burden resistor, R_22_, by the MOSFETs when they are activated.

The graph in Figure 22 illustrates the voltage across 1 Ω burden resistor R_22_ at 100 A (rms) when the MOSFETs are switched on. Again, each MOSFET (M_3_, M_4_) is expected to exhibit a resistance of approximately 3 mΩ when they are completely activated. At 100 A (rms) input current, the burden voltage can be seen as 0.50 V (rms).

As discussed in Section 6.2.2., the initial maximum value of burden voltage was not captured by the data acquisition card due to its measurement limitation. There is a possibility that initial burden voltage exceeds the LVT maximum voltage for a very short duration. TVS diode proposed to be connected in parallel to LVT provides protection for these high voltage spikes.

Again, the surges in the current graph indicate the actuation of the two MOSFET pairs within one fourth of the cycle. The rapid decrease in the burden voltage will be detected at the interrogator end of the actual sensor system.

#### 6.2.5. Investigations for the 1 Ω Burden Circuit for Different Phases of Input Current

Inevitably, by changing the phase of the input current, the activation time of the comparator output is affected. This is because the charging time of the capacitors within the regulator and threshold circuits is affected by the phase of the burden voltage, corresponding to the phase of the input current. It is expected that if a fault initiates at peak current of the input signal, the charging time of the circuit will be shorter than that for the fault initiation at a zero-crossing moment of the input signal. Following are some graphs that exhibit the behavior of the circuit by changing the phase of the input current at a magnitude of 60 A (rms) Figure 23, Figure 24 and Figure 25 illustrate the behavior of the 1 Ω burden circuit at different phases of the input current at a particular magnitude which in this case is 60 A (rms). As expected, it can be observed that when the phase angle increases from 0° to 90°, the activation time of the comparator decreases from 3.9 ms to 1.85 ms, respectively, and vice versa, when phase angle increases from 90° to 180° the activation time of the comparator increases from 1.85 ms to 3.9 ms. 

The graph in Figure 23 illustrates that as soon as the power from the Chroma source is turned on, at an input current of 60 A (rms) with 0° phase shift, the regulator output V_CC+_ saturates at 10.5 V at approximately 3.1 ms. The reference voltage V_ref_ saturates at 5.5 V at approximately 3.1 ms as well. The threshold detector voltage V_th_ lags V_ref_ initially. As soon as saturation of V_ref_ is achieved, V_th_ crosses V_ref_ at 3.5 ms. V_th_ after crossing V_ref_ saturates at 7.7 V at approximately 3.7 ms. As soon as V_th_ crosses V_ref_, V_out_ of the comparator turns high and saturates at 10.5 V at approximately 3.9 ms.

The graph in Figure 24 illustrates that as soon as the power from the Chroma source is turned on, at an input current of 60 A (rms) with 60° phase shift, the regulator output V_CC+_ saturates at 10.5 V at approximately 1.8 ms. The reference voltage V_ref_ saturates at 5.5 V at approximately 1.8 ms as well. The threshold detector voltage V_th_ lags V_ref_ initially. As soon as the saturation of V_ref_ is achieved, V_th_ crosses V_ref_ at 2.2 ms. V_th_ after crossing V_ref_ saturates at 7.7 V at approximately 2.4 ms. As soon as V_th_ crosses V_ref_, V_out_ of the comparator turns high and saturates at 10.5 V at approximately 2.56 ms.

The graph in Figure 25 illustrates that as soon as the power from the Chroma source is turned on, at an input current of 60 A (rms) with 90° phase shift, the regulator output V_CC+_ saturates at 10.5 V at approximately 1.3 ms. The reference voltage V_ref_ saturates at 5.5 V at approximately 1.3 ms as well. The threshold detector voltage V_th_ lags V_ref_ initially. As soon as the saturation of V_ref_ is achieved, V_th_ crosses V_ref_ at 1.4 ms. V_th_ after crossing V_ref_ saturates at 7.7 V at approximately 1.5 ms. As soon as V_th_ crosses V_ref_, V_out_ of the comparator turns high and saturates at 10.5 V at approximately 1.85 ms.

#### 6.2.6. Burden Voltage during the Intermediate Dip

As discussed in Section 2 and Section 3, the LVT maximum withstand voltage is 20 V rms. As discussed in Section 6.2.2., the burden voltage rises above the level of 10 V at certain instances, e.g., at the start of activation of MOSFETs and at the intermediate dip. An experimental investigation was performed to determine the maximum burden voltage level across the 10 Ω burden resistor. A voltage divider circuit was used to scale down the burden voltage across the 10 Ω burden circuit and was then again scaled up to its actual value to represent in the graph. In Figure 1 and Figure 5, it can be seen that TVS diode was placed in the original circuit to protect the LVT from overvoltage. It was therefore desirable to find if the voltage rise during the sudden deactivation of MOSFETs during the intermediate dip would breach the LVT safety threshold of 30 V. If the safety threshold was not breached, TVS diodes could be eliminated, simplifying circuit design. The graph in Figure 26 depicts the region of the intermediate dip for the 10 Ω burden circuit when a current of 90 A (rms) was applied. It is evident in the graph that the burden voltage is well below the range of the LVT maximum withstand voltage. The burden voltage suddenly rises within the LVT maximum withstand voltage level and then goes to minimum in less than 0.3 ms. Note that the actual spike may have been larger, but the temporal resolution of the data acquisition process would have prevented the visibility of the peak. Further investigation may be necessary to capture the signals around the peak with greater sampling frequency. 

## 7. Discussion

The behavior of the proof-of-concept autoranging circuit demonstrated through the series of experiments has been proven correct. When the input current was above 2 A, resulting in 22 V across the burden resistors (around the LVT limit), the 10 Ω burden circuitry activated while that of the 1 Ω burden circuit remained inactive, reducing the voltage across the burden (and the LVT) to 2 V, thus extending the measurement range. Furthermore, the results showed that when the current was increased beyond 20 A, the 1 Ω burden circuitry also activated, shorting the burden resistors with the internal on resistance of the MOSFET transistors. This would result with a voltage of around 1 V across both burden resistors and the LVT—enabling current monitoring but preventing the LVT and the burden resistors from damage due to excessive voltage/power dissipation.

It is clear from the graphs presented in the paper that the comparators function according to the design. The 1 Ω burden and 10 Ω burden comparator circuits are activated within the desired time and retain their high state continuously for over 1.5 s, followed by a fall in the output voltage of less than 1 ms. Within the 1ms, the circuitry recharges and the comparator high state is recovered, resulting in only a short-lasting voltage rise across the burden resistors. If this was to rise above the LVT damage threshold, the TVS diodes would activate to prevent the LVT from damage.

It was also observed that at the boundary of comparator activation, i.e., for the 10 Ω burden circuit at 2 A and 1 Ω burden circuit at 20 A, the activation time of the comparators is approximately 60 ms. This time reduces as the input current is increased and ultimately falls within the 5 ms window as the current increases. The extended inception time is acceptable at the boundary currents as the burden resistors and the LVT are designed to cope with these nominal currents on a continuous basis. 

It should be noted that the recording of the voltage/current signals from the control circuits for the 1 Ω and 10 Ω burden resistors were performed separately due to limitations of the data acquisition system (DAQ). However, the two parts of the circuit were combined and both functioning and performing their driving operations simultaneously, as evidenced by the surges in the current waveforms shown in Figure 15 and Figure 19. 

## 8. Conclusions and Future Work

In this paper a series of design steps and experimental investigations performed on the novel autoranging circuit are reported on. The results indicate that the novel approach can address an existing technological gap within the hybrid optical sensor by passively extending the measurement range of the device at its remote end. The paper has reported on a proof-of-concept two-stage autoranging circuit, its detailed design methodology, and extensive testing campaign. The circuit was based on high-performance MOSFETs acting as bi-directional switches. The experimental results were encouraging as the behavior of the proof-of-concept circuit was in accordance with the design. It was shown that the circuit correctly reacts to the emulated secondary CT current levels by bypassing the two burden resistors, 10 Ω and 1 Ω, at correct current thresholds. It was shown that this desired circuit behavior offers a passive, local sensitivity adjustment for the PCT device. It was also shown how such sensitivity adjustments can be detected by the remote interrogator through the measurement of the rate of change of the LVT voltage. The design of the MOSFET driver circuit was optimized through analytical calculations and fine-tuning of the component values during the experimental investigations. For example, the driver circuitry around the comparator is nearly optimal in terms of the power consumption as most of the energy accumulated during the charging phase is spent by the comparator during the high-state holding phase, minimizing the power surge during the charging phase. 

A further quantitative analysis will be carried out on the circuit design in the future. This will include the analysis of measurement errors during the MOSFET switching phase as the MOSFET driver drains power/current from the burden resistor which can be considered an error current. As stated above, this driver circuit can be further optimized in a way that it consumes least amount of power from the burden resistor. Further investigations of the autoranging circuit will be carried out with the real LVT and interrogator in the future. A switching detection algorithm will be deployed within the firmware of the interrogator. Furthermore, future experiments and investigations will involve primary injection and the utilization of actual CTs.

## Figures and Tables

**Figure 1 sensors-23-00551-f001:**
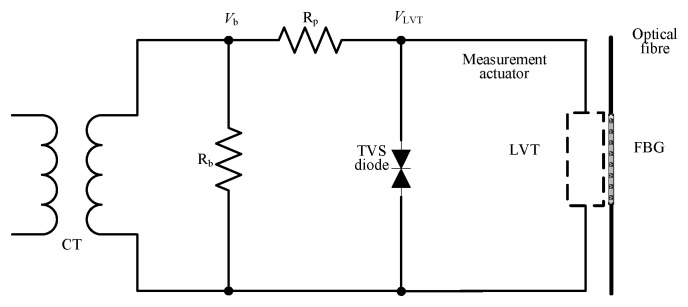
Electrical arrangement for connecting LVT to CT terminals.

**Figure 2 sensors-23-00551-f002:**
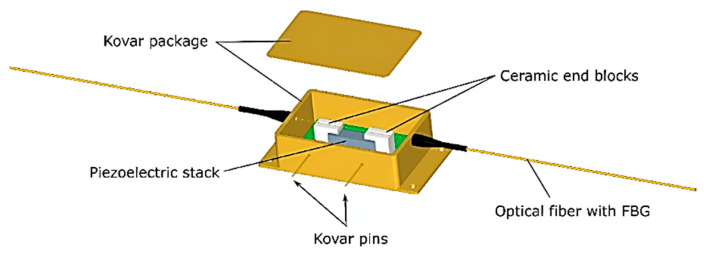
Low voltage transducer (LVT).

**Figure 3 sensors-23-00551-f003:**
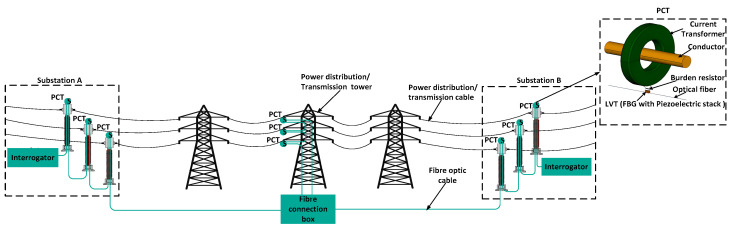
Example of PCT deployment at a power transmission/distribution system. An interrogator is conveniently located at a substation. Note that various other modes of deployment are possible as explained in detail in [8].

**Figure 4 sensors-23-00551-f004:**
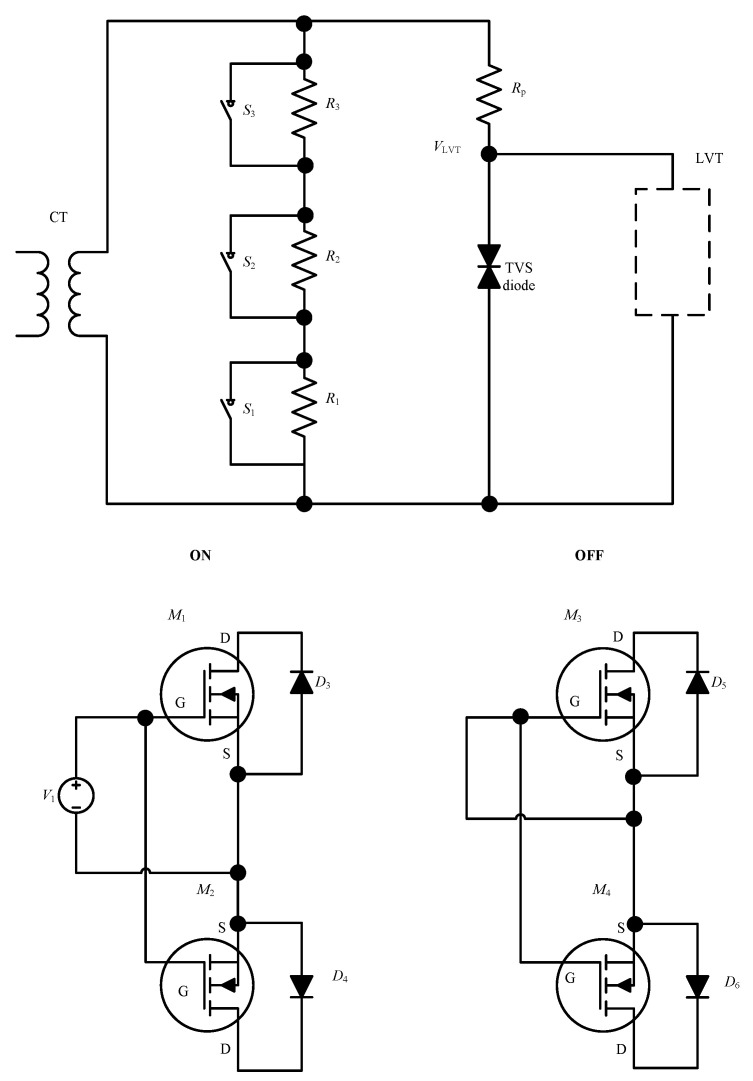
Illustration of the autoranging concept.

**Figure 5 sensors-23-00551-f005:**
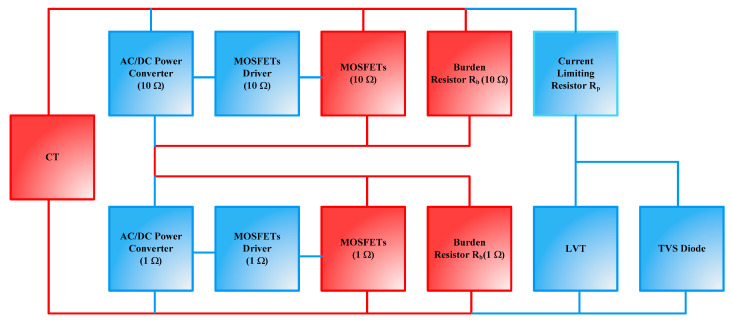
Block diagram of autoranging circuit for PCT.

**Figure 6 sensors-23-00551-f006:**
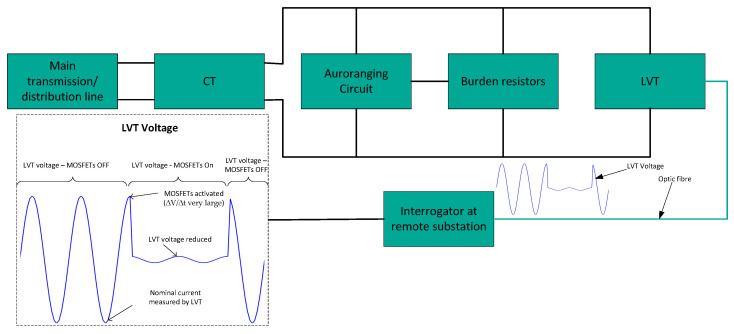
Example of range switching detected by interrogator.

**Figure 7 sensors-23-00551-f007:**
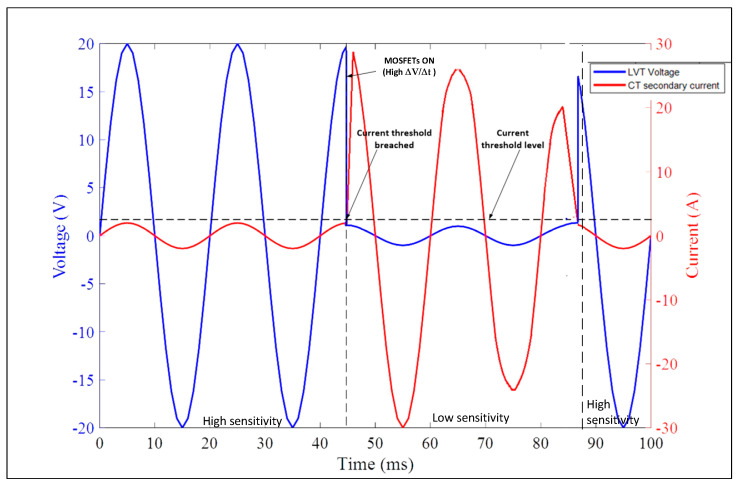
Example of fault current and corresponding LVT voltage signal. Note that the LVT voltage experiences a much greater rate of change at the point of current threshold level breach than the network fault current represented by the CT secondary current.

**Figure 8 sensors-23-00551-f008:**
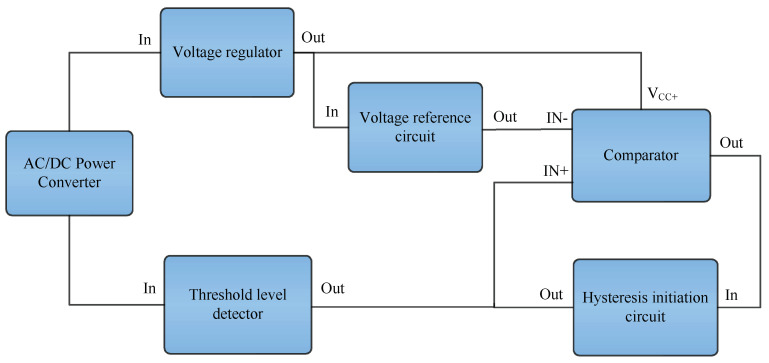
Block diagram of control circuit to drive MOSFETs.

**Figure 9 sensors-23-00551-f009:**
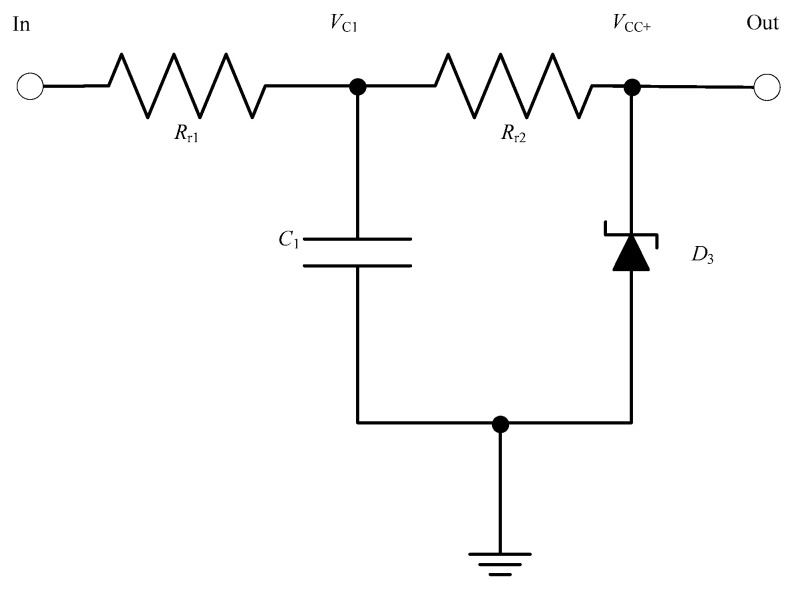
Circuit diagram of regulator.

**Figure 10 sensors-23-00551-f010:**
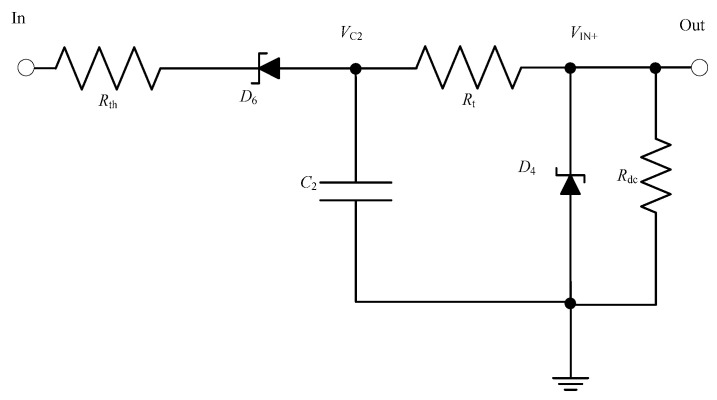
Circuit diagram of threshold level detector.

**Figure 11 sensors-23-00551-f011:**
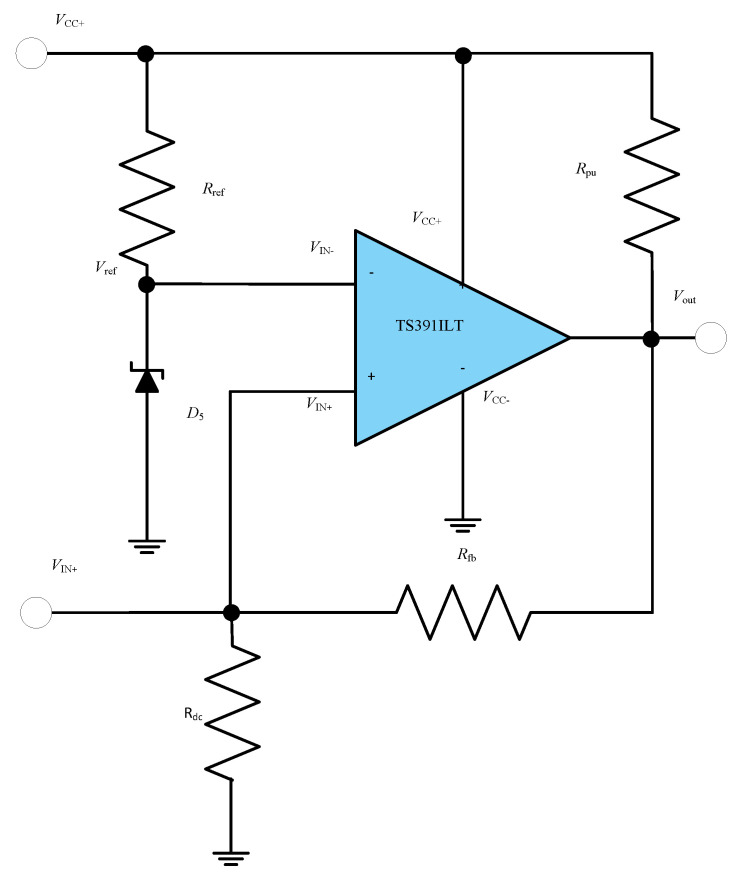
Circuit diagram of Schmitt Trigger Comparator.

**Figure 12 sensors-23-00551-f012:**
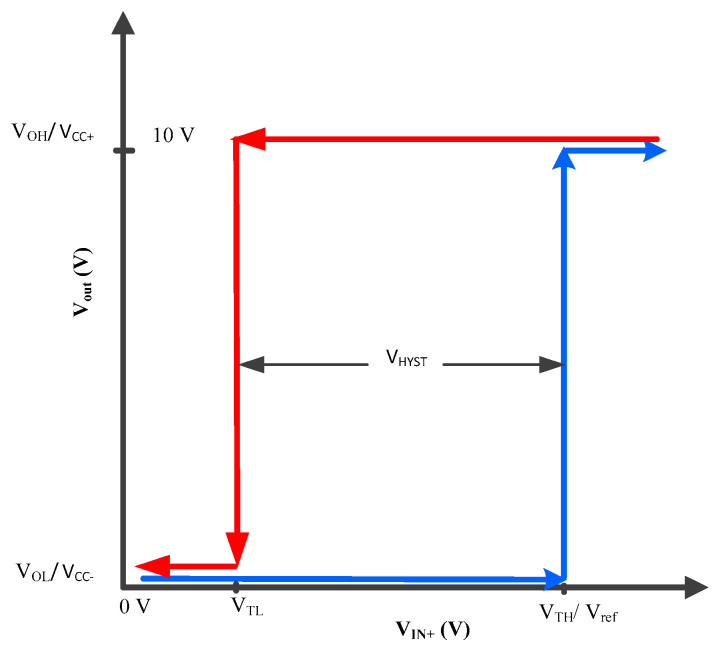
Hysteresis curve of comparator operation.

**Figure 13 sensors-23-00551-f013:**
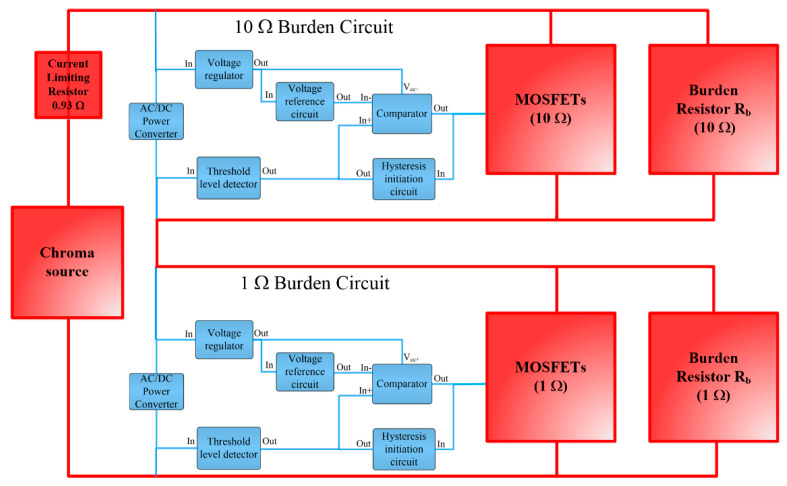
Block diagram of the secondary current injection test setup for the autoranging circuit.

**Figure 14 sensors-23-00551-f014:**
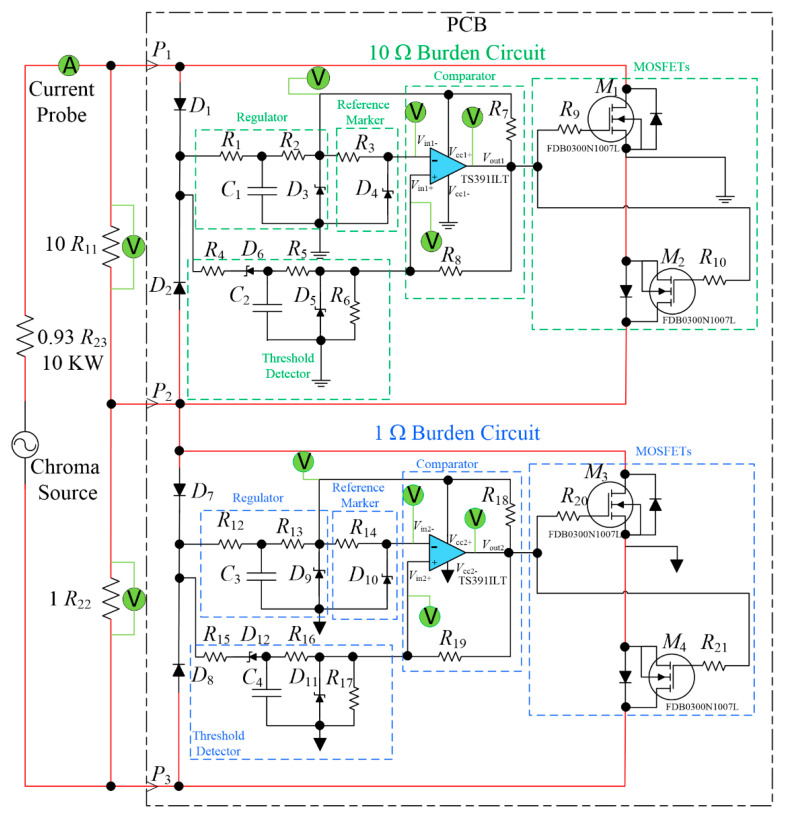
Component-level circuit diagram of the secondary current injection test set-up for the autoranging circuit.

**Figure 15 sensors-23-00551-f015:**
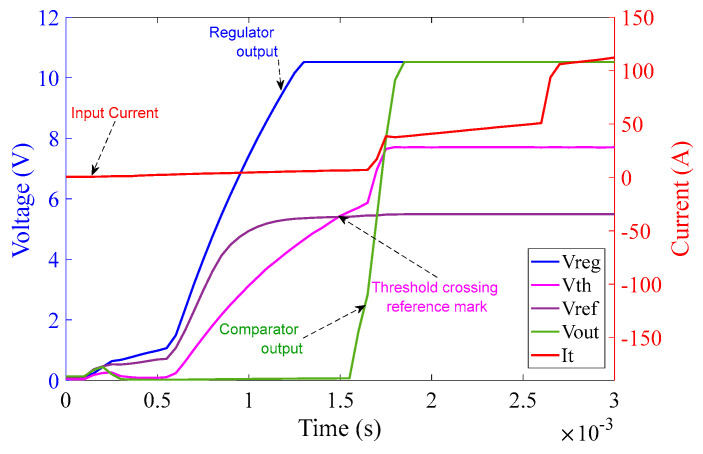
10 Ω burden circuit operation at 100 A (rms) input current over 3 ms.

**Figure 16 sensors-23-00551-f016:**
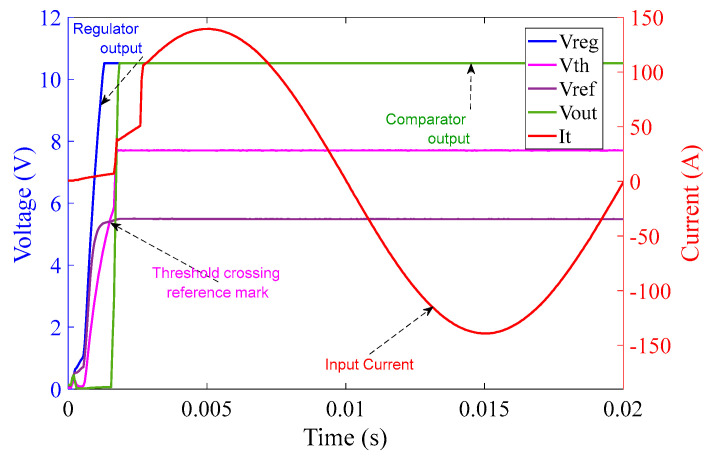
10 Ω burden circuit operation at 100 A (rms) input current over 20 ms.

**Figure 17 sensors-23-00551-f017:**
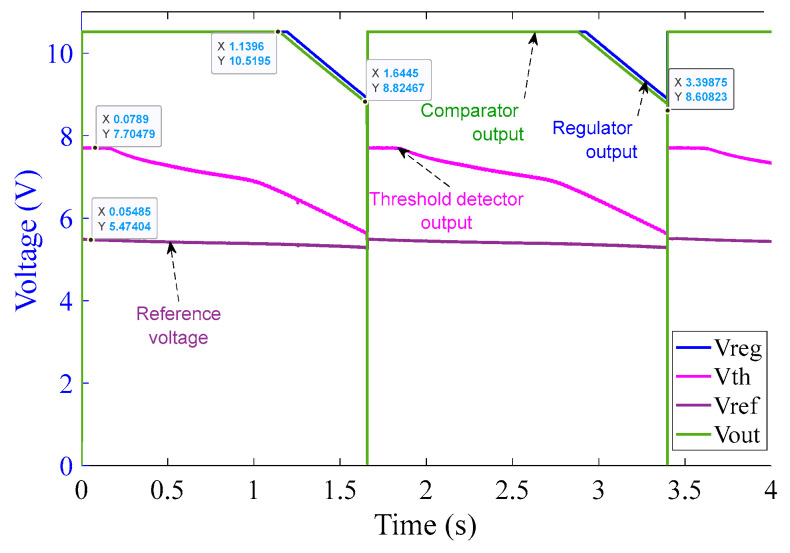
10 Ω burden circuit operation at 100 A (rms) input current over 4 s (input current not shown for clarity).

**Figure 18 sensors-23-00551-f018:**
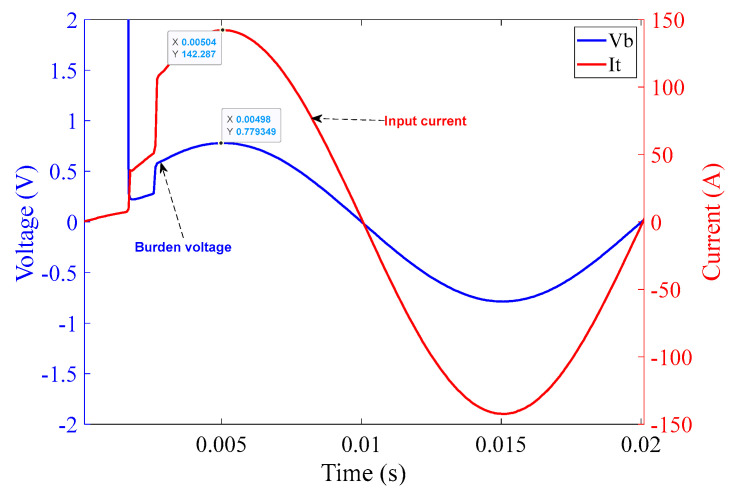
Burden voltage across the 10 Ω burden circuit at 100 A (rms) input current (MOSFETs switched on).

**Figure 19 sensors-23-00551-f019:**
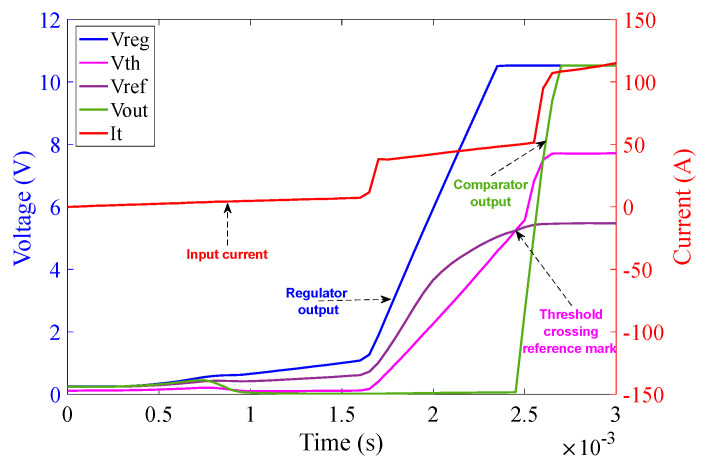
1 Ω burden circuit operation at 100 A (rms) input current over the duration of 3 ms.

**Figure 20 sensors-23-00551-f020:**
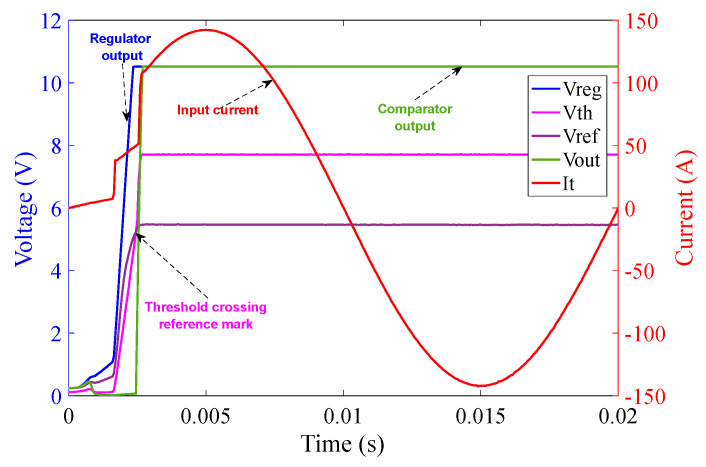
1 Ω burden circuit operation at 100 A (rms) input current over 20 ms.

**Figure 21 sensors-23-00551-f021:**
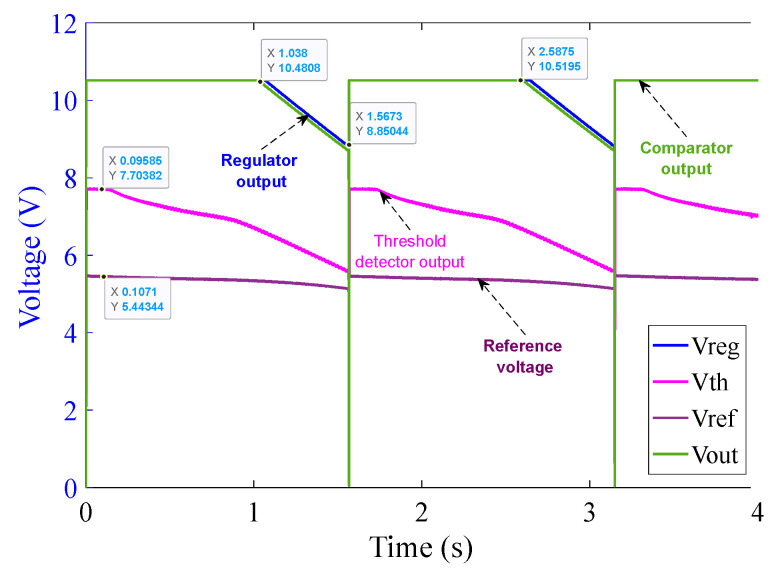
1 Ω burden circuit operation at 100 A (rms) input current over the duration of 4 s.

**Figure 22 sensors-23-00551-f022:**
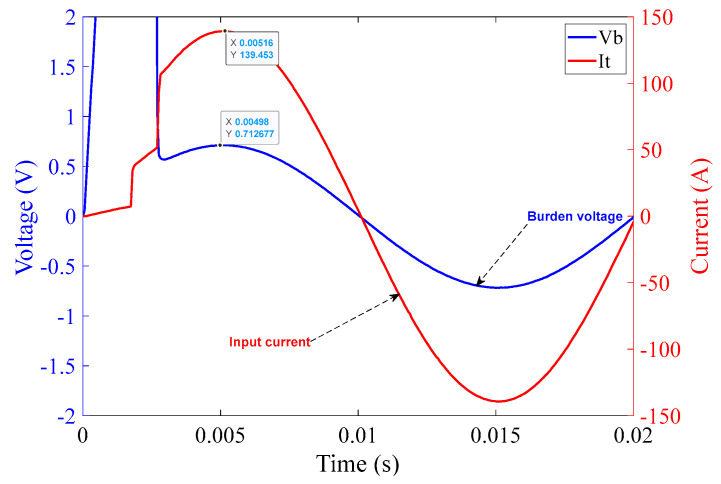
Burden voltage across 1 Ω burden circuit at 100 A (rms) input current.

**Figure 23 sensors-23-00551-f023:**
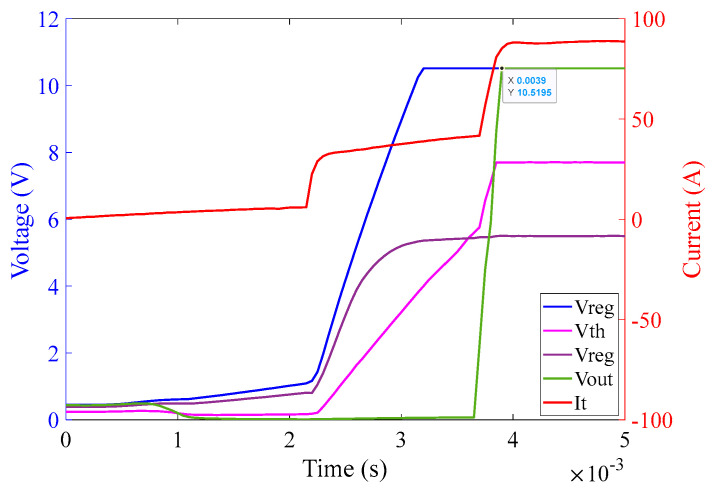
Activation time of the 1 Ω burden circuit at 60 A (rms) input current with 0° phase angle.

**Figure 24 sensors-23-00551-f024:**
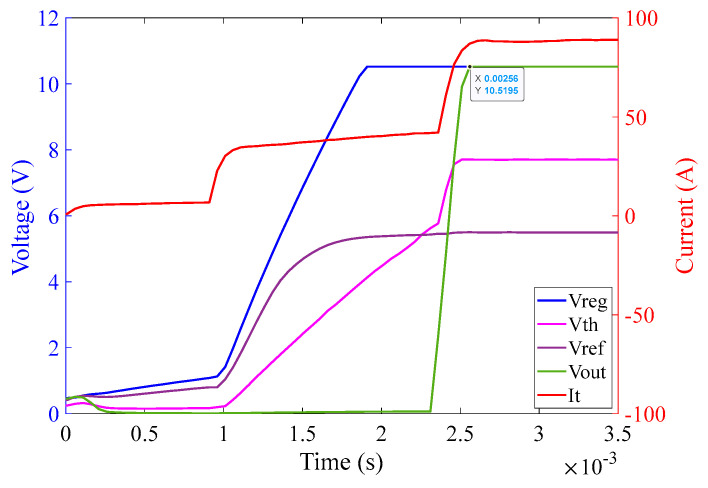
Activation time of the 1 Ω burden circuit at 60 A (rms) input current with 30° phase angle.

**Figure 25 sensors-23-00551-f025:**
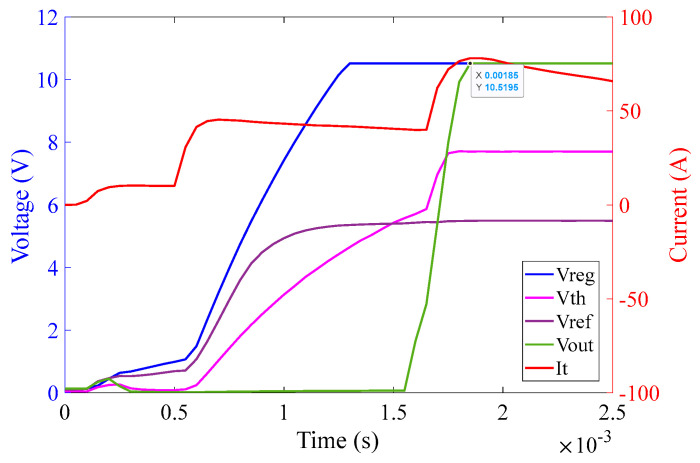
Activation time of the 1 Ω burden circuit at 60 A (rms) input current with 90° phase angle.

**Figure 26 sensors-23-00551-f026:**
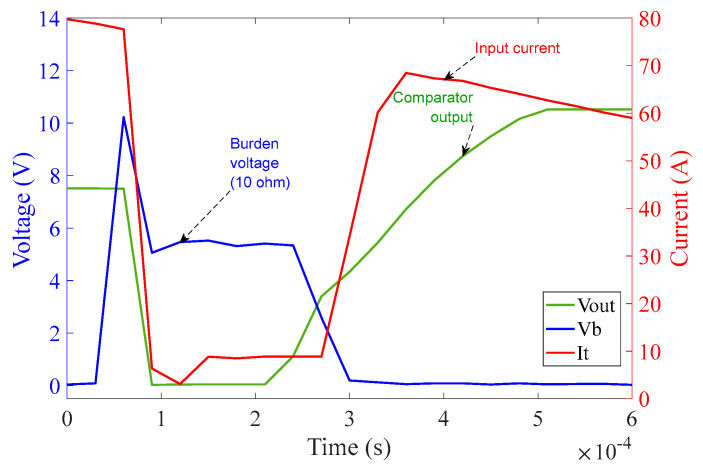
Burden voltage across 10 Ω burden circuit at 90 A (rms) input current at intermediate dip.

**Table 1 sensors-23-00551-t001:** IEC metering class specifications.

**Accuracy Class**	**±Percentage Current (Ratio) Error at Percentage of Rated Current Shown Below**				**±Phase Error at Percentage of Rated Current Shown Below**							
					**Minutes**				**Centiradians**			
	5	20	100	120	5	20	100	120	5	20	100	120
**0,1**	0,4	0,2	0,1	0,1	15	8	5	6	0,45	0,24	0,15	0,15
**0,2**	0,75	0,35	0,2	0,2	30	15	10	10	0,9	0,45	0,3	0,3
**0,5**	1,5	0,75	0,5	0,5	90	45	30	30	2,7	1,35	0,9	0,9
**1,0**	3,0	1,5	1,0	1,0	180	90	60	60	5,4	2,7	1,8	1,8

**Table 2 sensors-23-00551-t002:** IEC special application metering class specifications.

**Accuracy Class**	**±Percentage Current (Ratio) Error at Percentage of Rated Current Shown Below**					**±Phase Error at Percentage of Rated Current Shown Below**									
						**Minutes**					**Centiradians**				
	1	5	20	100	120	1	5	20	100	120	1	5	20	100	120
**0,2S**	0,75	0,35	0,2	0,2	0,2	30	15	10	10	10	0,9	0,45	0,3	0,3	0,3
**0,5S**	1,5	0,75	0,5	0,5	0,5	90	45	30	30	30	2,7	1,35	0,9	0,9	0,9

**Table 3 sensors-23-00551-t003:** IEC protection class specifications.

Accuracy Class	Current Error at Rated Primary Current %	Phase Error at Rated Primary Current		Composite Error at Rated Accuracy Limit Primary Current %	At Accuracy Limit Condition Maximum Peak Instantaneous Error %
		Minutes	Centiradians		
**5TPE**	±1	±60	±1,8	5	10
**5P**	±1	±60	±1,8	5	-
**10P**	3	-	-	10	-

**Table 4 sensors-23-00551-t004:** Description of the plot signals.

Variable	Description
** Vreg **	Regulator output voltage, V_cc+_
** Vth **	Threshold detector output voltage, V_in+_
** Vref **	Reference voltage, V_in−_
** It **	Total injected current from Chroma source
** Vout **	Comparator output voltage, V_GS_
** Vb **	Voltage across the burden resistor

## Data Availability

All data underpinning this publication are openly available from the University of Strathclyde KnowledgeBase at: https://doi.org/10.15129/18206014-b5a4-4b37-aef8-922803dc43ad, (accessed on 6 October 2022).

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
