# Peer review of "Design and Implementation of a Passive Autoranging Circuit for Hybrid FBG-PZT Photonic Current Transducer"

_sensors, 2023, doi:10.3390/s23010551_

Round 1

Reviewer 1 Report

In this paper, a passive auto ranging circuit for hybrid FBG-PZT photonic current transducer was designed and accomplished. The research included the description of the design process, construction and testing of a prototype passive auto ranging circuitry for integration with the PCT. The manuscript uses serially connected CT burden resistors and associated static MOSFET switches to realize instantaneous shortening of the resistors in response to increasing measured current. The two-stage circuitry is used to prove the concept and demonstrate the practically achievable circuit characteristics. A series of laboratory experiments were performed. Each of the parts was already presented by different groups (Circuit diagram of regulator, Circuit diagram of threshold level detector, Circuit diagram of Schmitt Trigger Comparator, the autoranging circuit, etc.)

The assembling of these ideas and the proof, is important. This is why I don’t reject this manuscript. While, there are some problems should be expressed and revised.

 1. The title of the article is not closely related to the content. From the title, the novelty here is the design and implementation of a passive autoranging circuit for hybrid FBG-PZT PCT, But there isnt a detailed description in the text about FBG and PCT for distributed measurement. In addition, the principle of passively autoranging based on photonic current transducer (PCT) is not clear.

2. The grounding resistance Rds is not displayed( connected with VIN+) in Figure 8.

3. The symbols used are confused in Figure 9,  VCC- and 0V appear at the same time.

4. There are plenty of format problems in this paper. The formats of figures, paragraphs and references should be in accord with the sample of the journal. The font sizes in figures should not larger than the text, such as Fig. 6, Fig. 7 and Fig. 8. The first line of a paragraph should be empty by two spaces. The format of references is confused, for example reference 20 and reference 22.

5.  This paper is limited to the description of the design process, construction and testing of a prototype passive auto ranging circuitry for integration with the PCT. So the paper is incomplete. Readers do not clearly know the problems the research needs to solve.

6. To overcome drawbacks of the traditional optical NCITs, novel photonic transducers utilizing fiber Bragg grating (FBG) sensors and piezoelectric (PZT) components were proposed. While, in this paper FBG only was mentioned. Most of the energy was devoted to researching the structure of input circuit. This is a technical problem, not a theoretical problem.

7. In part 6, experiments have been done to verify auto ranging concept. The two parts of the circuit (associated with the 10Ω and 1Ω burden resistors) were investigated separately. There is no comparison with relevant research. So the feasibility of this auto ranging structure can not be verified.

8. The voltage and current records were used to determine the overall characteristics of the circuit along with the switching behavior of the MOSFETs. The input current and voltage of this structure are relatively low. How can it be applied in the power transmission system?

 The research has technical value; however, the paper is lack of enough proof to confirm the conclusion. It is short of innovation in this paper. So suggest that the paper should be modified and reviewed again.

Reviewer 2 Report

The MS highlights a autoranging circuit. It needs certain aspects to be checked again.

a) The inroduction part seems a little bit disorirnted.

b) The no. of references are very small. Claims are to be endorsed by recent references.

c) Conclusion part has to be in line with the findings. Future aspect need not be so much highlighted.

d) Figure 15 to Figure 20 are to be explained in more depth.

e) How does VGS appear in Opam ST circuit?

f) So many circuits are described. Hwoever, the linkages are missing fully.

g) Discussion part has to be more extensive in tune with recent published work.

Round 2

Reviewer 1 Report

The manuscript is focused on a novel technique for passively autoranging a photonic current transducer (PCT) . The idea of using FBG technology is exciting. The usage of single-mode fiber and FBG, multiple PCTs can be interconnected and distributed over a long distance. The relies on the usage of multiple, serially connected CT burden resistors and associated static MOSFET switches to realize instantaneous shortening of the resistors in response to increasing measured current. The autoranging technique proposed in this article can realize passively, utilizing a modular, μW-power comparator circuit that powers itself from the electrical energy supplied by the CT within a small fraction of the 50/60 Hz cycle. The work is useful in the field of sensing. The authors have done an interesting piece of work for the scientific community working and I recommend this manuscript for acceptance.

Reviewer 2 Report

It is now acceptable.